DOI: 10.1038/s41467-018-05046-2　　**OPEN**

# Tools for engineering coordinated system behaviour in synthetic microbial consortia

Nicolas Kylilis [1,2], Zoltan A. Tuza[1,2], Guy-Bart Stan [1,2] & Karen M. Polizzi [2,3]

Advancing synthetic biology to the multicellular level requires the development of multiple cell-to-cell communication channels that propagate information with minimal signal interference. The development of quorum-sensing devices, the cornerstone technology for building microbial communities with coordinated system behaviour, has largely focused on cognate acyl-homoserine lactone (AHL)/transcription factor pairs, while the use of non-cognate pairs as a design feature has received limited attention. Here, we demonstrate a large library of AHL-receiver devices, with all cognate and non-cognate chemical signal interactions quantified, and we develop a software tool that automatically selects orthogonal communication channels. We use this approach to identify up to four orthogonal channels in silico, and experimentally demonstrate the simultaneous use of three channels in co-culture. The development of multiple non-interfering cell-to-cell communication channels is an enabling step that facilitates the design of synthetic consortia for applications including distributed bio-computation, increased bioprocess efficiency, cell specialisation and spatial organisation.

---

[1] Department of Bioengineering, Imperial College London, London SW7 2AZ, UK. [2] Imperial College Centre for Synthetic Biology (IC-CSynB), Imperial College London, London SW7 2AZ, UK. [3] Department of Life Sciences, Imperial College London, London SW7 2AZ, UK. Correspondence and requests for materials should be addressed to G.-B.S. (email: g.stan@imperial.ac.uk) or to K.M.P. (email: k.polizzi@imperial.ac.uk)

Synthetic biology research and applications to date have mostly been focused on the engineering of homogeneous or monoclonal designer cell populations to perform functions ranging from bio-computation[1], to bioproduction of biomaterials[2] and chemicals[3], and biosensing[4,5] among others. De novo implementation of genetic circuits of increasing complexity and size in living cells is an important challenge in synthetic biology. Obstacles include unwanted interactions between genetic parts[6], restrictions on the number of available genetic parts[1], limits to the size of DNA that can be transformed into the host cell and metabolic burden to the host chassis[7]. The use of consortia of organisms could potentially alleviate these bottlenecks provided that the functionality of the genetic circuit can be distributed among different cell populations[8]. Additionally, the use of synthetic consortia could expand the capabilities and applications of synthetic biology by enabling compartmentalisation[9], cell specialisation[10], parallel bio-computation[11], increased bioprocess efficiency[12] and spatial organisation[13].

A vital component for the engineering of synthetic consortia is devices that enable communication between the different populations to coordinate their behaviour. Quorum-sensing systems, especially those based on small molecules such as acyl-homoserine lactones (AHLs), have become the favoured technology for engineering cell-to-cell communication[14] because of their simple genetic architecture. AHLs are produced enzymatically by the expression of a single enzyme, such as the acyl-homoserine-lactone synthase protein that is the product of the *luxI* gene[15]. AHL molecules can freely diffuse in the intracellular and extracellular environment. Intracellularly, AHLs bind transcription factor proteins, which results in an activated complex that can bind a specific quorum-sensing promoter to initiate transcription of downstream genes[16]. The most frequently used quorum-sensing parts, the LuxR protein and its cognate $P_{lux}$ promoter have been assembled into a AHL-receiver device that has been extensively characterised in terms of its input–output response, dynamic performance upon induction, specificity to other AHL molecules, and evolutionary reliability[17]. Various studies have sought to increase the number of available AHL communication modules with functional devices built from the biological components of the las[18], tra[18], rpa[18], rhl[19], cin[19] and esa[20] quorum-sensing systems. Nevertheless, quorum-sensing devices can exhibit various degrees of crosstalk either in the form of AHL molecules binding to non-cognate transcription factors (chemical crosstalk) or transcription factors binding to non-cognate promoters (genetic crosstalk)[13]. The degree of orthogonality between designed AHL communication modules can be quantified by high-throughput screening as demonstrated for a set of designed AHL-receiver devices of the lux, las, rpa and tra quorum systems[18]. For this set of devices only the tra and rpa devices were determined to be completely orthogonal for both chemical and genetic crosstalk[18]. However, a number of strategies have been demonstrated for minimising crosstalk, e.g. modulating the expression levels of the transcription factor that influence the response of the device or quorum-sensing promoter engineering[13].

In this research, we aim to increase the number of available tools for the engineering of microbial communities. Initially, we design and construct AHL-receiver devices from genetic components of the rhl, lux, tra, las, cin and rpa quorum-sensing systems and characterise their input–output behaviour at the individual cell level. Next, we characterise the functionality of these AHL-receiver devices in the presence of non-cognate AHL inducers. In doing so, we create, to the best of our knowledge, the largest characterised library of quorum-sensing devices with mapped chemical crosstalk interactions. The size of this database and the emerging combinatorial complexity hinder manual search for possible communication channels, and thus calls for an automated process. The fact that this database contains data about all possible combinations of AHL-receiver devices and AHL inducers enables us to create a software tool to automatically select orthogonal AHL communication channels. This automated process facilitates the design of synthetic microbial consortia. We demonstrate the power of this approach by experimentally validating one of the algorithmically proposed designs for the specific control of gene expression through three non-interfering AHL communication channels in a polyclonal *E. coli* co-culture. The large number of cell-to-cell communication devices built and characterised here in combination with the developed software tool for automatic identification of non-interfering communication channels facilitate the design of synthetic microbial communities with multiplex interaction engineered to allow the development of applications in biotechnology, ecology and medical intervention.

## Results

**Design and characterisation of AHL-receiver devices.** To expand the pool of well-characterised cell-to-cell communication tools for engineering microbial consortia, synthetic AHL-receiver devices were constructed, which employed components from six quorum-sensing systems (Fig. 1). The engineered systems incorporated genetic elements from the lux system[21] (*Vibrio fisheri*), the rhl[22,23] and las[24,25] systems (*Pseudomonas aeruginosa*), the cin[26] system (*Rhizobium leguminosarum*), the tra[27] system (*Agrobacterium tumefaciens*) and the rpa[28] system (*Rhodopseudomonas palustris*). The cognate AHL inducer signal molecule for each system is indicated in Fig. 1a. The AHL-receiver devices share an identical genetic architecture and were cloned into the pSB1C3 vector backbone for propagation in cells (Fig. 1b). For in vivo characterisation of the AHL-receiver devices, a GFP reporter module was cloned downstream of the quorum-sensing promoters and cells were transformed with the resulting composite device plasmids. The transformed cells were cultured, induced using the cognate AHL, and the resulting cell fluorescence was assayed by flow cytometry. All the AHL-receiver devices functioned as intended, as confirmed by the increase in cell fluorescence at higher AHL inducer concentrations (Fig. 2a). GFP expression output was homogeneous for all AHL inducer concentrations despite differences in the tail length of inducer molecules and the presence of additional functional groups that may affect the permeability of the cell membrane to the inducer[29]. Many applications in synthetic biology require careful modulation of intracellular gene expression levels in response to inducer concentrations. However, a number of inducible systems frequently used in genetic circuit design exhibit an on or off response at the individual cell level despite the appearance of a graded response at the population level[30]. The six devices characterised here avoid this behaviour and can therefore be used to tune genetic expression levels over a wide range.

The input–output function of the AHL-receiver devices was determined by fitting the data (Fig. 2b) with a four-parameter logistical curve model (Eq. 1):

$$\text{GFP}([\text{AHL}]) = b + \frac{a - b}{1 + 10^{((\log(\text{EC50}) - \log([\text{AHL}])) * h)}} \quad (1)$$

where $a$ is the maximal GFP output, $b$ is the basal GFP output, [AHL] is the AHL inducer concentration, GFP is the green fluorescence response of the device for that inducer concentration, EC50 is the inducer concentration that results in half maximal activation of the device, and $h$ is proportional to the value of the steepest slope along the curve (Hill coefficient) that indicates the responsiveness of the device to the input. To facilitate the

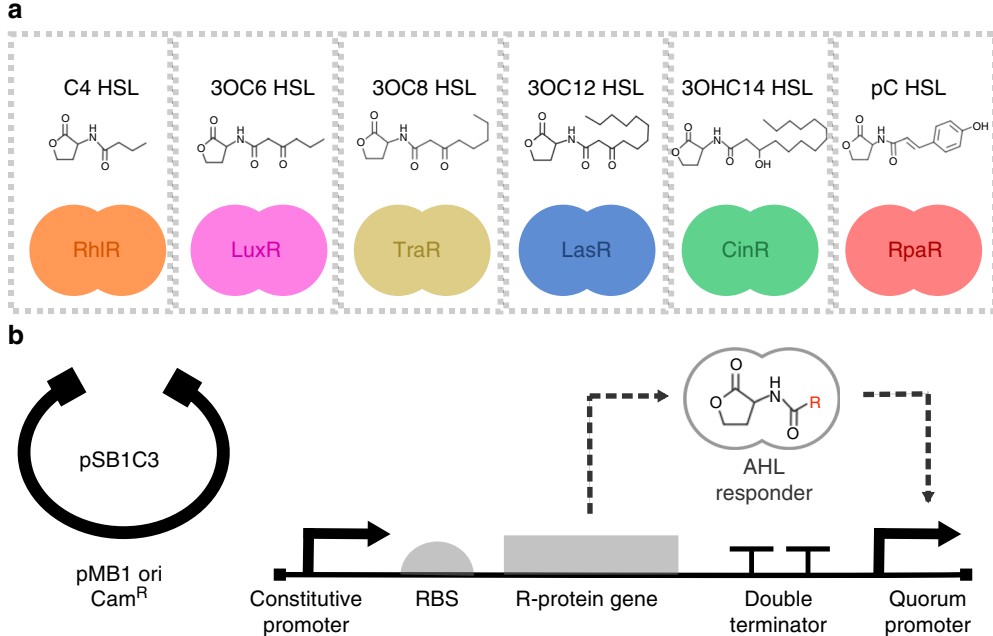

**Fig. 1** Library of AHL-receiver devices. **a** Quorum-sensing systems with schematics of the transcription factor regulator protein and chemical structures of its cognate AHL ligand. **b** Vector backbone and genetic architecture of AHL-receiver devices. The devices were constructed using the following genetic elements from the Registry of Standard Biological Parts: Bba_J23101 (constitutive promoter), Bba_B0034 (RBS), Bba_B0015 (double terminator)

computer-aided design of genetic circuits for engineering microbial consortia when using these devices, the fitted parameter values for each device are provided in Supplementary Table 4. Additionally, Supplementary Note 1 discusses use-cases for the metrics derived, and Supplementary Figure 1 displays fold change and normalised EC50 curves of cognate pairs of AHL-receiver devices/AHL inducer for use in genetic circuit design.

**Chemical signal crosstalk of non-cognate devices**. Next, we sought to investigate the chemical signal crosstalk of the AHL-receiver devices as this can severely limit the engineering of synthetic consortia that require the use of multiple communication channels. The characterisation of the GFP output for all six AHL-receiver devices was carried out as described above for each of the 5 non-cognate AHL inducers, thus creating the largest library of AHL-receiver devices with all their chemical signal interactions quantified. The experimental results are presented in Fig. 2c in the form of heat maps of normalised GFP output for each device.

These results show that each device exhibited a distinct profile of crosstalk interactions, and that all devices were effectively activated by non-cognate AHL molecules. Amongst the devices, the rpa system exhibited the least propensity for activation by non-cognate AHL molecules in comparison to its cognate AHL inducer. This behaviour likely stems from the distinct molecular structure of the p-coumaroyl HSL inducer compared to the rest of the AHL molecules (Fig. 1a). Despite the apparent chemical crosstalk, it was clear that the use of particular combinations of AHL-receiver devices operating within restricted AHL inducer concentration regimes could allow the simultaneous use of multiple communication channels.

**Relative activity of devices to facilitate integration**. To facilitate the re-use of the AHL-receiver devices demonstrated here in combination with other synthetic biology devices and systems, the basal and maximal expression levels for each combination of

AHL-receiver device and inducer were calibrated against a reference promoter. Similar to previous work[31], the in vivo activity of the J23101 constitutive promoter from the Anderson library was set as the reference standard with a relative promoter activity of 1 (Supplementary Note 2 & Supplementary Figure 2), and expression characteristics of the engineered AHL-receiver devices were calibrated according to this reference. The engineered systems were determined to vary widely both in relative basal and maximal expression strength (Fig. 3 & Supplementary Table 1). This is an important consideration for their integration with downstream modules in engineered systems. Furthermore, their documented, varying relative expression strength provides a broad design space for genetic circuit design.

**Automated identification of orthogonal communication**. To facilitate the engineering of microbial consortia, we developed a software tool that automatically identifies combinations of AHL-receiver devices and inducers that behave orthogonally within a given inducer concentration regime. As an input, the software uses the database of fitted model parameters for the AHL-receiver devices with cognate and non-cognate AHLs. Then, the software identifies orthogonal communication channels, i.e. suitable combinations of AHL-receiver devices and AHL inducer concentrations that satisfy user-defined specifications such as i) minimal activation thresholds for specific gene expression, ii) maximal activation thresholds for non-specific gene expression and iii) number of simultaneous chemical communication channels required. At the core of the software tool is an algorithm that searches for orthogonal chemical channels by treating this problem as an instance of the combinatorial Rook problem (the algorithm and the technical analysis of the software implementation are further discussed in Supplementary Note 3, Supplementary Figures 2–5 and Supplementary Table 2).

To demonstrate the utility of this software tool, we used it to identify suitable systems that would allow for a number of chemical communication systems with minimal signal crosstalk

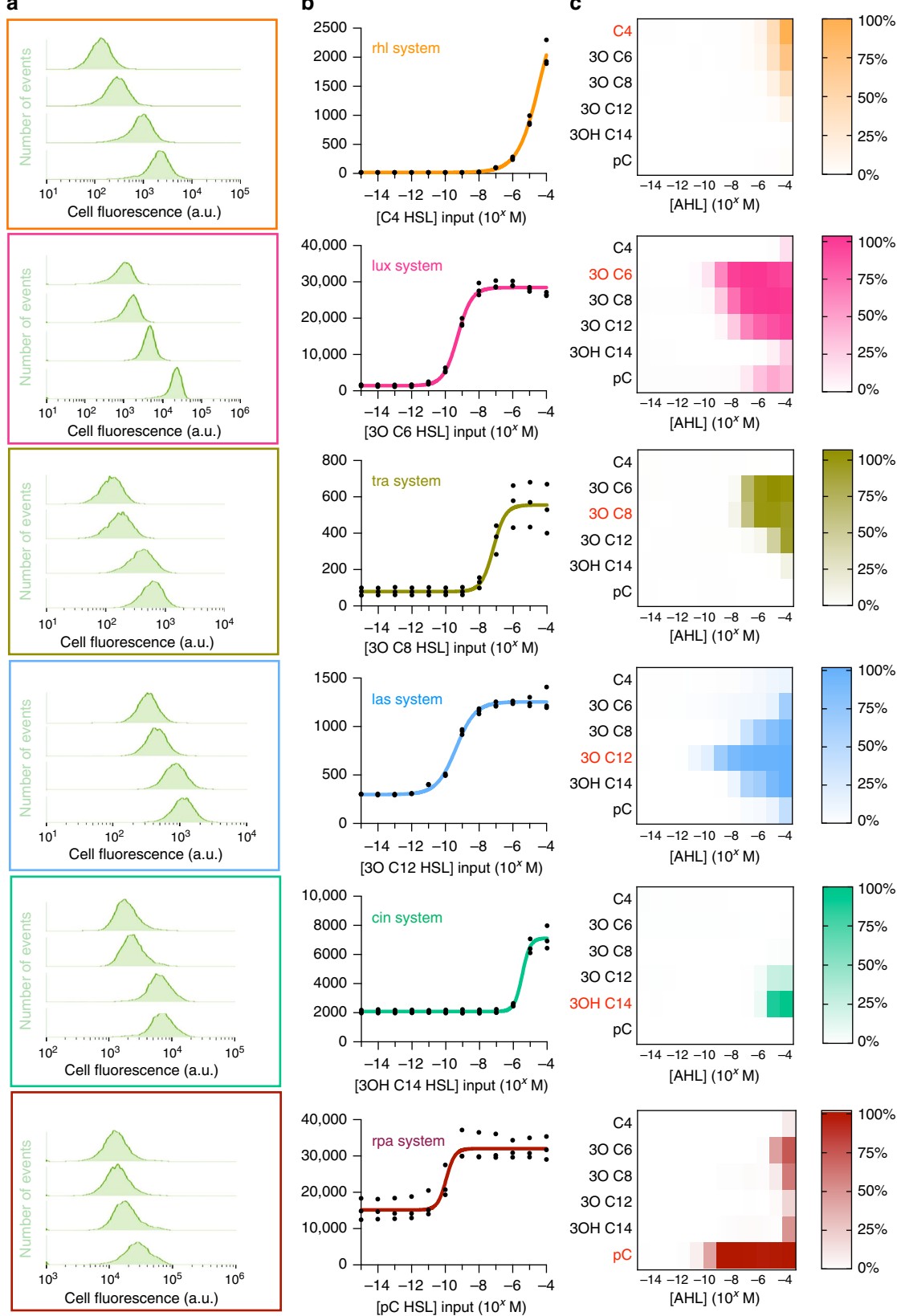

**Fig. 2** Characterisation of AHL-receiver devices. **a** Histograms of cell fluorescence from flow cytometry data of AHL-receiver composite devices/AHL inducers cognate pairs at selected inducer concentrations. **b** Input–output functions of AHL-receiver devices, i.e. GFP output in arbitrary units (au) against cognate AHL inducer concentrations derived when mean fluorescence data were fitted with a four-parameter logistical function model (Eq. 1). Points represent individual data for three biological replicates and coloured lines represent the fitted logistical model. **c** Heat maps of GFP output from AHL-receiver composite devices induced with cognate (coloured red) and non-cognate AHL inducer concentrations. The GFP output was normalised against the maximal expression of the cognate system. Coloured frames and plots correspond to AHL-receiver devices as follows (from top to bottom): rhl system (orange), lux system (pink), tra system (yellow-green), las system (light blue), cin system (green) and rpa system (red)

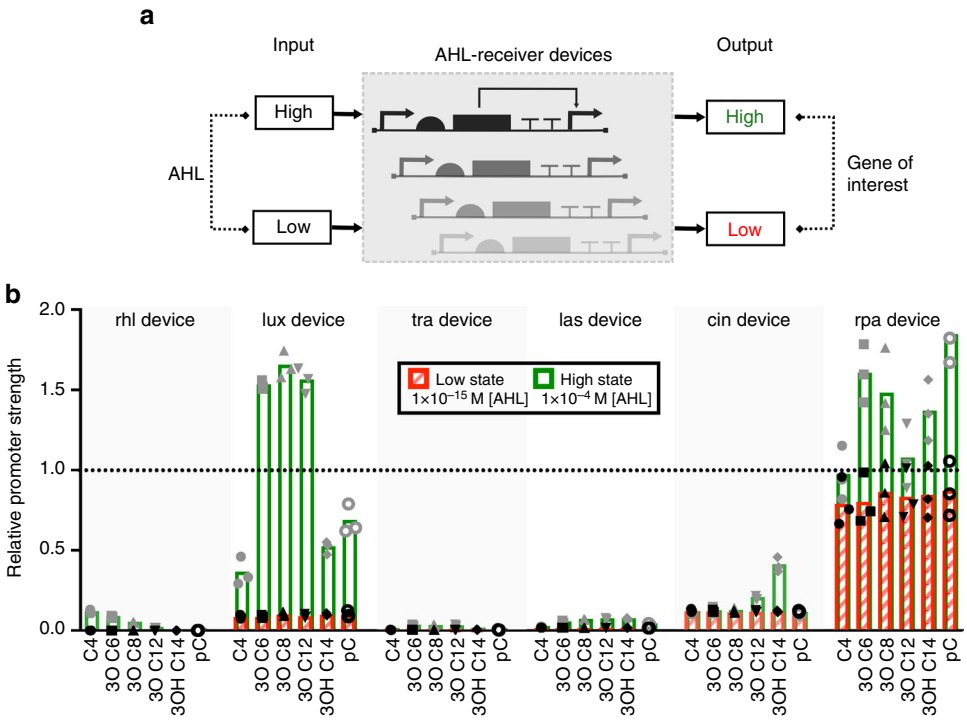

**Fig. 3** Relative promoter strengths of AHL-receiver devices. **a** Schematic of possible states of characterised genetic devices. AHL-receiver devices can operate in two states: at low input they exhibit a low output; while at high input they exhibit high output. **b** Bar chart of relative promoter strength of inducible AHL-receiver devices for each AHL inducer calibrated against the output of the reference standard promoter (Bba_J23101). The low state values were derived from GFP fluorescence data at $1 \times 10^{-15}$ M AHL, while the high state values were derived from GFP fluorescence data at $1 \times 10^{-4}$ M AHL induction. The dotted line indicates output of the reference standard promoter Bba_J23101. Individual points indicate calculated relative strength values from three biological replicates and the bar height indicates their mean value

in a hypothetical engineered consortium. Initial design specifications focused on the identification of two quorum-sensing communication channels that would allow for more than two-fold activation of specific gene expression, while exhibiting less than two-fold non-specific activation of gene expression (crosstalk). The software was able to identify several systems that met the specifications. Among the identified systems were pairs of AHL-receiver device/AHL inducer that utilised their cognate AHL inducers (Fig. 4a, top), systems of pairs that utilised non-cognate AHL inducers (Fig. 4a, middle) and systems of pairs that utilised a combination of cognate and non-cognate AHL inducers (Fig. 4a, bottom). In a further use-case example, we set stricter input specifications for the specific activation threshold of more than ten-fold and a crosstalk threshold of less than three-fold across two communication channels. The software tool identified a system consisting of the rhl device with C4 HSL in the $1 \times 10^{-6}$–$1 \times 10^{-5}$ M concentration regime and the lux device with 3 O C6 HSL in the $1 \times 10^{-9}$–$1 \times 10^{-7}$ M concentration regime (Fig. 4b) that meets these pre-defined orthogonality criteria. Furthermore, in other use cases the software tool was able to identify several systems for the simultaneous use of three

orthogonal chemical communication channels (an example is presented in Fig. 4c), and even a system of four channels with minimal signal crosstalk (Fig. 4c). In the latter two use cases the gene expression activation thresholds had been set at more than two-fold-specific activation and less than two-fold non-specific activation.

To experimentally validate the predictive capabilities of this software tool, we implemented a system of three orthogonal communication channels; this particular design is presented in Fig. 4c (top panel). The system suggested by the software tool was implemented in genetically modified *E. coli* strains using the rhl, lux and las composite devices that incorporate a GFP reporter module. The engineered strains were also transformed with plasmids encoding constitutive expression of mOrange, mRFP1 or mtagBFP2 fluorescent protein to enable bacterial strain differentiation when in co-culture (Fig. 4d, 3d-plot). The co-culture system was tested for the presence of orthogonal communication channels by the addition C4 HSL [$1 \times 10^{-5}$M], 3 O C8 HSL [$1 \times 10^{-8}$M] and 3OHC14 HSL [$1 \times 10^{-7}$M] in all single and dual input combinations with the change in cell fluorescence recorded by flow cytometry. The experimental

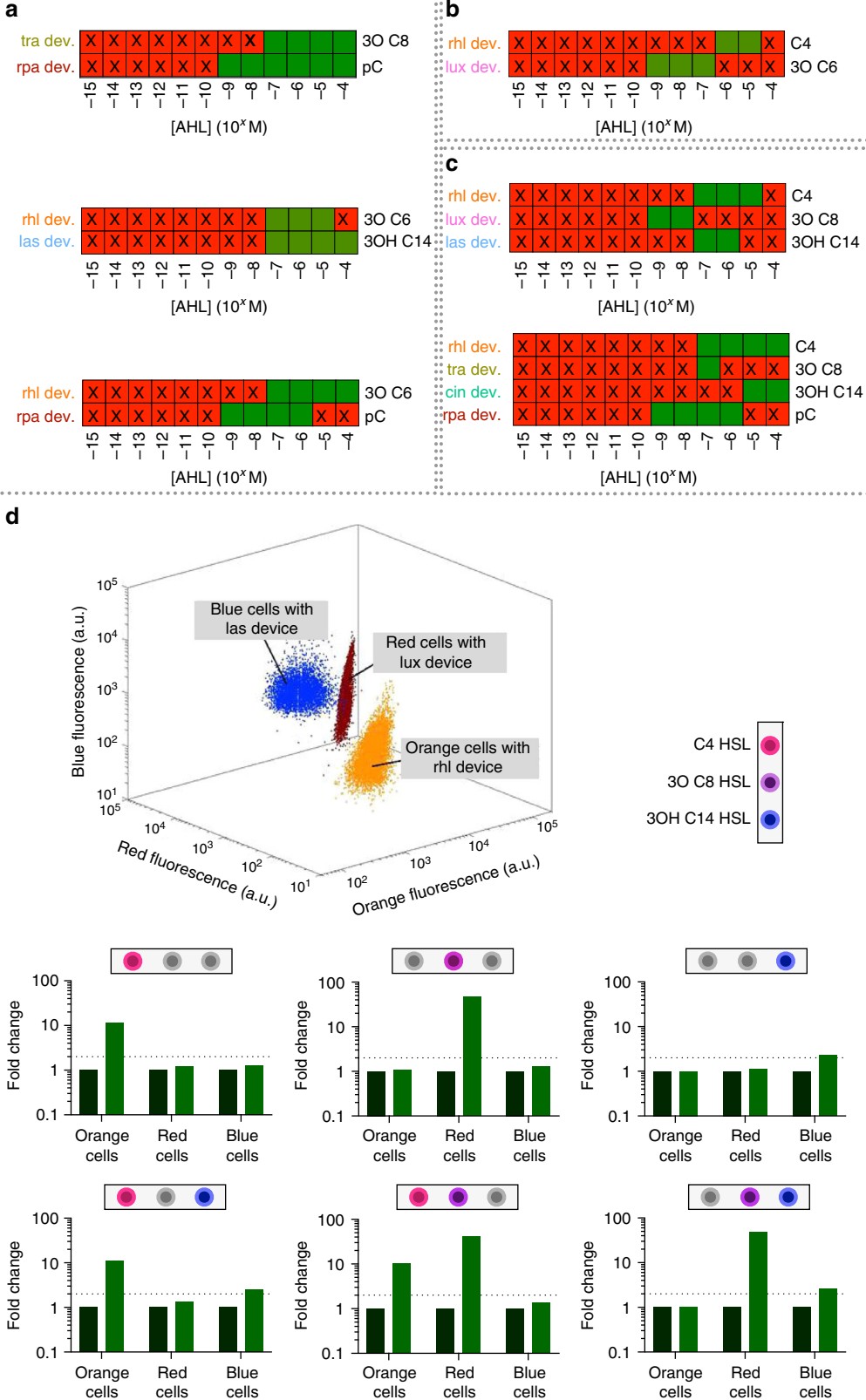

results showed that the calculated fold change in GFP output was: (i) more than two-fold for specific gene expression and (ii) less than two-fold for non-specific gene activation across all three chemical communication channels (Fig. 4d). These results were in agreement with the in silico predictions and demonstrated the benefit of this software as a computer-aided-design tool.

## Discussion

In this work, we describe the construction of a large library of AHL-receiver devices. This library consists of six AHL-receiver devices constructed using genetic parts mined from various microbial species. These devices were experimentally characterised in terms of their response to cognate and non-cognate

**Fig. 4** Computer-aided design of synthetic microbial consortia with orthogonal communication channels. **a**–**c** Orthogonal AHL-chemical communication channels as identified by our computer-aided-design software. For each panel, crossed red indicates an AHL inducer concentration which does not meet at least one of the user-defined specifications, while green indicates an AHL inducer concentration for which all user-defined specifications are met. **a** Systems of two communication channels designed with the following specifications: two-fold-specific gene activation and less than two-fold crosstalk. **b** System of two communication channels designed with the following specifications: more than ten-fold specific gene activation and less than three-fold crosstalk. **c** System of three communication channels (top) and four communication channels (bottom) designed with the following specifications: more than two-fold-specific gene activation and less than two-fold crosstalk. **d** Co-culture of three *E. coli* populations forming a system that is predicted to allow for orthogonal control of gene expression by the use of three chemical channels (as proposed in **c**: top illustration); Individual cell populations in co-culture were differentiated by flow cytometry by recording cell fluorescence resulting from mOrange (orange cell population), mRFP1 (red cell population) or mtagBFP2 (blue cell population) expression (3D-pot). The calculated fold change in GFP output for each cell population in response to chemical induction is indicated in the bar charts below the 3D-plot. The dotted line in the bar charts represents the user-specified threshold value for gene activation and crosstalk signal. Derived fold change values were calculated from one experimental implementation of the co-culture

AHL inducer molecules. These input–output responses were then parametrised in terms of their basal expression, maximal expression, fold activation, input signal *EC50* concentration for half-activation and device sensitivity. Such metrics are important in assessing the suitability of the device for use in genetic circuit designs. For example, the fold activation metric can be of significance when incorporating these devices with downstream processes that require overcoming specific thresholds for activation or inhibition, such as in logic gate designs[1,32]. At the same time, the *EC50* value can be of importance for engineering bacterial populations that respond to cell population densities[33].

Additionally, both cognate and non-cognate pairs of AHL-receiver devices and AHL inducers have been shown to be able to activate gene expression at different levels of signal output. This behaviour, termed as chemical crosstalk, can complicate the design of synthetic microbial consortia. Here, through the characterisation of a total of 36 synthetic quorum systems (cognate and non-cognate pairs) and the development and use of a computer-aided design tool, we were able to use the signal crosstalk as a design feature to identify ranges of AHL inducer concentrations that delineate orthogonal chemical communication. This enables the simultaneous use of our AHL-receiver devices for the engineering of microbial consortia. We used our approach to identify a potential system using up to four orthogonal channels simultaneously. Furthermore, we experimentally validated this approach by engineering a polyclonal co-culture capable of controlling gene expression using three non-interfering AHL communication channels.

The software tool (source code is provided as Supplementary Software 1) is designed to aid the identification of a user-specified number of orthogonal communication channels from a library of characterised AHL-receiver devices. This computer-aided-design software can flexibly accommodate user-defined constraints for fold changes in gene expression and identify any number of desired communication channels, which can be particularly useful in enabling consortia designs of increasing complexity as the number of characterised quorum systems expands.

To fully realise the potential of engineered microbial consortia, additional research could provide methodologies and design rules to further expand the chemical design space for orthogonal communication and control of gene expression in multicellular systems. Beyond effective cell-to-cell communication technologies, another important aspect in consortia engineering is the robust maintenance of cell populations in co-cultures in order to ensure the long-term co-existence of different engineered species. Recent developments that set the theoretical foundations for such designs include the use of negative feedback architectures to robustly control gene expression in multicellular systems[34] and the use of control systems for setting specific ratios of cell types in polyclonal multicellular systems[35]. Development of effective design strategies that enable the rational engineering of synthetic microbial consortia will lead to novel application areas for synthetic biology[36], and further tools for use in basic research to understand natural ecosystems[37,38].

It is expected that the ability to rationally design synthetic consortia and repurposing the enormously diverse catalytic biochemistry of microorganisms will allow significant gains in bioprocess efficiency for biomass re-utilisation. Currently this has been demonstrated for ethanol production from cellulose[10] or xylan[39] feedstocks among other examples[36,40]. In addition, consortia of microorganisms can provide elegant and efficient solutions for bioenergy production from renewable resources as is the case for prototroph/heterotroph cell factories[41]. Other foreseeable applications of cell consortia will probably take advantage of the parallel bio-computation enabled by molecular compartmentalisation to interrogate multiple biomarkers[42] simultaneously. First applications may apply to areas such as microbiome engineering and next-generation medical diagnostics[43], and then interface with human physiology to effect change during pathology via integrated genetic logic[44,45]. Finally, the current efforts in engineering microbial consortia will provide a better understanding of multicellular systems behaviour and will develop foundational technologies for future applications in tissue and/or organ engineering[46,47] that will transform cell-based therapeutic interventions.

## Methods

**Biological constructs and chemicals**. Experiments were carried out with *E. coli* TOP10 cells transformed with plasmid constructs (Supplementary Table 3). Plasmid constructs were assembled to comply with the BioBrick RCF [10] standard using a variety of molecular cloning techniques. Biological parts were acquired from the registry of Standard Biological parts or from chemical synthesis. DNA sequences of assembled AHL-receiver devices are available for download (10.5281/zenodo.1252305). AHL inducers utilised were purchased as follows: N-Butyryl-DL-homoserine lactone {C4 HSL} (09945 Sigma), 3-oxohexanoyl-L-homoserine lactone {3OC6 HSL} (K3007 Aldrich), N-(3-Oxooctanoyl)-L-homoserine lactone {3OC8 HSL} (O1764 Sigma), N-(3-Oxododecanoyl)-L-homoserine lactone {3OC12 HSL} (O9139 Sigma), N-(3-Hydroxytetradecanoyl)-DL-homoserine lactone {3OHC14 HSL} (51481 Sigma) and N-(p-Coumaroyl)-L-homoserine lactone {pC HSL} (07077 Sigma). AHLs stock solutions were made in 100% DMSO.

**Cell cultures and AHL induction**. *E. coli* TOP10 cells transformed with plasmid constructs were cultured for in vivo GFP expression measurements as follows: Overnight culture of transformed cells were diluted 1:100 into LB medium supplemented with 34 µg/mL chloramphenicol and grown for 3 h at 37 °C with shaking at 250 rpm. Cultures were diluted to an $OD_{600}$ of 0.1 in chloramphenicol supplemented LB medium and 100 µL transferred into 96-well flat-bottom microplates. The wells were supplemented with AHL inducer (1:100 dilution) at appropriate concentrations and the cultures grown for an additional 3 h in a shaking incubator at 37 °C and 750 rpm before measurement.

**Data analysis**. Samples from cultured *E. coli* cells transformed with the plasmid constructs for the composite devices were analysed by flow cytometry (Supplementary Methods). Data analysis was carried out using GraphPad Prism version 7.0®. For each of the AHL-receiver device, the mean cell fluorescence was calculated from three biological repeats. To determine GFP output, the mean cell

autofluorescence value of *E. coli* TOP10 cells was subtracted from the mean cell fluorescence of cells transformed with the AHL-receiver composite devices. The resulting values were fitted with a four-parameter logistical curve model (Eq. 1). The fitting of the model was constrained with basal expression GFP output values derived from samples treated with $1 \times 10^{-15}$ M inducer concentration of the cognate AHL inducer. Fold activation for each device was calculated by dividing the maximal GFP output by the basal GFP output from the parameter values obtained from the fitted model, where the devices showed GFP output saturation. Otherwise, fold activation was determined by the use of the GFP output value at 100 μM inducer concentration (highest soluble concentration under the experimental conditions). Relative activity of AHL-receiver devices was calculated by normalising their GFP output to the GFP output of the J23101 promoter. The same methodology was used to determine the relative activity of members of the Anderson library of constitutive promoters.

**Co-culture assay**. Single colonies of *E. coli* TOP10 cells transformed with two plasmid constructs (a quorum-sensing composite device with a GFP reporter module, and a reporter plasmid that constitutively expressed another fluorescent protein as described in main text) were cultured overnight in LB medium supplemented with chloramphenicol (34 μg/ml) and ampicillin (50 μg/ml) as described above. A spectrophotometer was used to determine the $OD_{600}$ value of cell cultures. This information was used to dilute cell cultures to an $OD_{600}$ of 0.1 in antibiotic supplemented LB medium, and these were then added together as a co-culture. A volume of 100 μl co-culture was transferred into 96-well flat-bottom microplate wells and induced with AHL at appropriate concentrations as defined in the main text. Finally, the microplate co-cultures were grown for 3 h in a shaking incubator at 37 °C and 750 rpm. Co-culture samples were analysed using a BD LSRFOR-TESSA X-20 flow cytometer (BD Biosciences). Initially, cell populations were gated for blue fluorescence (excitation (ex): VL405nm/emission (em): VL450/50) and orange-red fluorescence (ex: YGL488nm/em: YGL586/15), and subsequently the orange-red population was gated for orange (ex: YGL561nm/em: YGL586/15) and red fluorescence (ex: YGL561nm/em: YGL670/30). The gated populations were subsequently analysed for green cell fluorescence (ex: BL488nm/em: YGL525/20). Analysis of mean cell fluorescence data for fold changes in GFP output was carried out as described above. Analysis of flow cytometry data was carried out using the FlowJo™ v10 software.

**The software**. The implementation of the algorithm for the automatic identification of orthogonal chemical channels was developed in MATLAB® from MathWorks. The flowchart of the algorithm is illustrated in Supplementary Figure 5. The source code and the database of AHL-receiver device transfer function parameters can be found in Supplementary Software 1. The software has been designed to be compatible with databases that respect the provided data structure. Green-red colour matrices illustrating the solutions identified by the software were drawn using GraphPad Prism® v7.

**Code availability**. The source code for the current version of the software (ChemmComms-calculator-v1.0) and any future versions will be available at: https://github.com/GBS-SynBioLab/ChemComms-calculator.

**Data availability**. The data that support the findings of this study are available from the corresponding authors upon request and can be used without restriction.

DNA sequences of AHL-receiver devices have been deposited in Zenodo with the following accession code: 10.5281/zenodo.1252305.

The parameters of fitted models for the AHL-receiver devices have been deposited in Zenodo with the following accession code: 10.5281/zenodo.1252276.

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

## Acknowledgements

We acknowledge the following funding bodies for support: the Engineering and Physical Sciences Research Council (EPSRC) - EP/P009352/1 (NK, GBS), EP/M002187/1 (GBS), EP/K038648/1 (KP) and the EU H2020-FETOPEN-2016-2017, project 766840, COSY-BIO (ZAT, GBS).

## Author contributions

Study design: N.K., G.B.S. and K.M.P. Execution of experiments: N.K. Data analysis: N.K. Algorithm development: N.K., Z.A.T. Writing manuscript: N.K. Editing manuscript: G.B.S., K.M.P., N.K., Z.A.T.

## Additional information

**Competing interests:** The authors declare no competing interests.

