## [Peer Review File · Nature Communications]

Reviewers' comments:

Reviewer #1 (Remarks to the Author):

In the manuscript "Tools for engineering coordinated system behaviour in synthetic microbial consortia" by Dr. Polizzi and colleagues, the authors have constructed a library of six quorum sensing signals that respond to different input quorum sensing signals (AHLs), and have mapped the cross-signal interactions. They have characterized these library in terms of the specificity of each circuit in response to the different AHLs (their orthogonality) and determined the promoter strengths in response to different concentrations of the different signals. They have shown that the different systems have different levels of specificity when comparing cognate versus non-cognate signals, and while some are very specific and only respond to the cognate signal others respond to a broader range of signals at different concentrations. They also show that the promoters have different sensitivity and provide easy and accessible data for future users to choose the best promoters for each particular need in respect to specificity and promoter strength. Additionally, and importantly, they also showed that these strains can be used to construct engineered microbial consortia of multiple-species. For that they mixed three strains, expressing different promoters and showed that these constructs can be used to manipulate communities to express promoters at different levels in response to different signals.

I think this report provides a good collection of promoters which can be very useful for the implementation of synthetic genetic circuits important for the implementation of designed-base synthetic consortia. These can then be used in a variety of synthetic biology applications. Engineered QS-regulated circuits have been explored with the aim of constructing synthetic circuits, but those usually focused on the ability of expressing one promoter at the time or different promoters in series. The important novelty here is that the authors provide the tools and the information required to implement communities that can express different promoters in response to signal inputs with different specificities.

I think the work is well performed and the data is very useful. I have only some important suggestions for improving the figures and the figures legends. In general, I thought that the figures/legends were lacking important details which are needed to fully understand the data. Below I list several suggestions which I hope will help to improve the manuscript.

Minor Comments:

1 – Figure 1B shows two regions B0034 and B0015, but then these regions are not mentioned neither in the text nor the figure legend, so it is not clear what they are. Figure legend should refer to the meaning of JS23101.

2 – Figure 2A, the y axis needs to be defined. Figure 2B, I am assuming that these 6 plots correspond to the 6 promoters with the different color codes, but these should be explained in the figure legend.

3 – Figure 2C, here it would be useful if the cognate signal for each circuit would be highlighted (for example underlined) in each plot.

4 – Figure 3C, this figure is not well explained. What are the different bars for each of the circuits, response to the 6 different signals? Which concentrations of signals were used here?

5 – The authors mention the Anderson library in Figure 3A, but don't discuss it on the paper. I think it should be discussed in the paper. I think the illustration of the Anderson library should be removed from figure 3A. This is not relevant to show it as the setup is very simple. It should be discussed in the paper instead.

Reviewer #2 (Remarks to the Author):

This paper investigates multiple cell-to-cell communication channels. In contrast to other work already in the literature, a systematic classification is presented of a large library of AHL-receiver devices whose input/output behaviour is characterized experimentally. The need for orthogonal communication channels and the classification proposed in the paper is in my view an important problem and therefore I believe the paper to be timely and relevant, particularly in the domain of

synthetic biology.

The result of the analysis which uses both experiments and some mathematical modelling is encoded in a software tool that the authors propose for automated identification of orthogonal chemical communication channels and is made available as supplementary information to the manuscript.

I found the paper clear and well written. From a methodological viewpoint the authors tackle a problem of relevance particularly, as they recognize in the paper, for the current effort in Synthetic Biology aimed at constructing multicellular consortia. A striking example is that of multicellular control systems which the authors refer to on line 203 although without any further detail. (I would suggest adding a reference to recent work on this interesting topic.)

The experimental work is well described and the mathematical models reasonable for the cognate and non-cognate pairs under investigation.

I would have liked to see more details on the software tool. For example, when the user specifies the conditions that need to be met what guarantee is there that a pair can be found that matches? Also typically how long it takes for the software to "converge" onto a solution? It might be that given the limited number of combinations to be examined the software works well on average but a more quantitative assessment of its performance and the underlying algorithm would enrich the paper if added to the supplementary info.

As a final consideration, I believe this is a methodological paper whose findings can be useful in several domains such as those identified in the manuscript. It is not going "to influence thinking in the field" but will certainly be a useful addition to the design and analysis tools available in the literature. I am therefore left wondering whether it might better fit a more specialist journal in the area of computational biology.

Letter response to reviewers' comments on manuscript: NCOMMS-17-33832, "Tools for engineering coordinated system behaviour in synthetic microbial consortia", **Nicolas Kylilis, Zoltan A. Tuza, Guy-Bart Stan and Karen Polizzi**

We thank the reviewers for taking the time to provide detailed feedback on our paper. We hope the responses detailed below will clarify any questions and will satisfy the concerns that were raised.

Reviewer 1:

Comment #1 – Figure 1B shows two regions B0034 and B0015, but then these regions are not mentioned neither in the text nor the figure legend, so it is not clear what they are. Figure legend should refer to the meaning of JS23101.

Response: We agree with the reviewers' comment that the Biobrick genetic part identification code used in the figure is not something that the average reader of the manuscript will be familiar with. To address this, we have now edited the figure so that the genetic construct includes generic descriptions of its constituent parts that should be familiar to most researchers in biological sciences. For example, the B0015 part has been re-labelled as a double terminator, the B0034 part has been renamed RBS etc. Additionally, we have added a segment in the figure caption that provides the available identification codes (as they would appear in the Registry of Standard Biological Parts) for each genetic element.

Comment #2 – Figure 2A, the y axis needs to be defined. Figure 2B, I am assuming that these 6 plots correspond to the 6 promoters with the different colour codes, but these should be explained in the figure legend.

Response: In Figure 2A, the y-axis of the histograms has been now defined as the "number of events" to denote the type of data derived from flow cytometry experiments. In agreement with the suggestion regarding Figure 2B, a segment has been added to the figure caption explaining the colour codes used in the figure and their relation to the appropriate quorum sensing systems.

Comment #3 – Figure 2C, here it would be useful if the cognate signal for each circuit would be highlighted (for example underlined) in each plot.

Response: The cognate signal has now been labelled in red to indicate that this inducer is the cognate one for the particular AHL-receiver device. A segment has been added to the figure caption to explain this convention.

Comment #4 – Figure 3C, this figure is not well explained. What are the different bars for each of the circuits, response to the 6 different signals? Which concentrations of signals were used here?

Response: In accordance to the suggestions, the bar chart in this figure now includes labels that indicate that each bar is associated with a particular pair of AHL-receiver device / AHL inducer. Additionally, the figure caption now includes a segment that states that the values provided for high/low states correspond to the experimentally derived GFP output for induction at 1×10^{-15} M [AHL] and 1×10^{-4} M [AHL], respectively.

Comment #5 – The authors mention the Anderson library in Figure 3A, but don't discuss it on the paper. I think it should be discussed in the paper. I think the illustration of the Anderson library should be removed from figure 3A. This is not relevant to show it as the setup is very simple. It should be discussed in the paper instead.

Response: As suggested by the reviewer, the illustration for the Anderson promoter library has now been removed from Figure 3A. Additionally, after further consideration regarding the message we wanted to deliver to the reader in this figure, we removed the bar chart for the Anderson promoter library characterisation. Instead we now supply these data in the supplementary information (Figure S2 and Table S3). Also, as suggested by the reviewer, we provide a more extended discussion on the Anderson promoter and the reasons why we carried out the characterisation of the mentioned promoters in the supplementary information (Page 7, lines 30-46).

Reviewer 2:

Comment #1: I found the paper clear and well written. From a methodological viewpoint the authors tackle a problem of relevance particularly, as they recognize in the paper, for the current effort in Synthetic Biology aimed at constructing multicellular consortia. A striking example is that of multicellular control systems which the authors refer to on line 203 although without any further detail. (I would suggest adding a reference to recent work on this interesting topic.)

Response: We thank the reviewer for the insight into the topic and suggestion. We have now added lines 217-220 that cite two recent applications of control feedback architectures to control gene expression and cell-type ratios in multicellular systems.

Comment #2: I would have liked to see more details on the software tool. For example, when the user specifies the conditions that need to be met what guarantee is there that a pair can be found that matches? Also typically how long it takes for the software to "converge" onto a solution?

It might be that given the limited number of combinations to be examined the software works well on average but a more quantitative assessment of its performance and the underlying algorithm would enrich the paper if added to the supplementary info.

Response: Taking into account the reviewer's comment, we have revised our algorithm which now computes possible solutions in polynomial time. This replaces our previous implementation that was running in exponential time. The structure of the new algorithm and an in-depth analysis of its computational complexity and some example run times are now available in the supplementary information document (Pages 8-11). We are confident that the new implementation with the presented complexity characteristics will serve the community in the future even with a much larger of combinations of pairs available for analysis.

Regarding convergence, as illustrated in the flowchart of the software structure (Figure S5), the software tool examines all possible solutions, and if no solution that satisfies user-defined specifications is available the algorithm terminates and displays a message to the user that "no solution was found to achieve the stated specifications". The user will then be

able to “relax” their specifications to attempt to find a feasible solution given these new user-defined specifications.

Comment #3: As a final consideration, I believe this is a methodological paper whose findings can be useful in several domains such as those identified in the manuscript. It is not going "to influence thinking in the field" but will certainly be a useful addition to the design and analysis tools available in the literature. I am therefore left wondering whether it might better fit a more specialist journal in the area of computational biology.

Response: We believe *Nature Communications* is the ideal outlet for our work because it will enable it to have wide visibility among our target audiences of synthetic biologists, biotechnologists, and bioengineers. We think our work, encompassing a large amount of new experimental data and the associated software tool (that was designed for the non-programmatically inclined end-user), will be useful to many researchers—equally for biological sciences researchers and mathematical modellers—who want to use quorum sensing in the design of synthetic systems. We suspect that submitting this work to a more specialist journal will make it much less accessible and visible to the broad sets of researchers and end-users interested in the currently hot topic of microbial consortia engineering, e.g. synthetic biologist, biotechnologists, and bioengineers.

REVIEWERS' COMMENTS:

Reviewer #1 (Remarks to the Author):

The reviewers have addressed all my concerns satisfactorily. As I wrote in my initial assessment this paper provides a good collection of promoters which can be very useful for the implementation of synthetic genetic circuits important for the implementation of designed-base synthetic consortia.

Reviewer #2 (Remarks to the Author):

I thank the authors for taking carefully into account my previous comments. I believe the paper is interesting and deserves publication. Re-reading the manuscript I concur with the authors that their work combining experiments and software development can be of interest to the broad readership of Nature Communications.